# Feasible Classified Models for Parkinson Disease from ^99m^Tc-TRODAT-1 SPECT Imaging

**DOI:** 10.3390/s19071740

**Published:** 2019-04-11

**Authors:** Shih-Yen Hsu, Hsin-Chieh Lin, Tai-Been Chen, Wei-Chang Du, Yun-Hsuan Hsu, Yi-Chen Wu, Po-Wei Tu, Yung-Hui Huang, Huei-Yung Chen

**Affiliations:** 1Department of Information Engineering, I-Shou University, No.1, Sec. 1, Syuecheng Rd., Dashu District, Kaohsiung City 84001, Taiwan; h.shihyen@gmail.com (S.-Y.H.); wcdu@isu.edu.tw (W.-C.D.); 2Department of Chinese Medicine, E-Da Cancer Hospital, No.1, Yida Rd., Jiaosu Village, Yanchao District, Kaohsiung City 82445, Taiwan; blueloveecru@hotmail.com; 3School of Chinese Medicine for Post-Baccalaureate, I-Shou University, No.8, Yida Rd., Jiaosu Village, Yanchao District, Kaohsiung City 82445, Taiwan; 4Department of Medical Imaging and Radiological Science, I-Shou University, No.8, Yida Rd., Jiaosu Village, Yanchao District, Kaohsiung City 82445, Taiwan; ctb@isu.edu.tw (T.-B.C.); yhhuang@isu.edu.tw (Y.-H.H.); 5Department of Nuclear Medicine, E-Da Hospital, I-Shou University, No.1, Yida Rd, Jiaosu Village, Yanchao District, Kaohsiung City 82445, Taiwan; edps900002@edah.org.tw (Y.-H.H.); ed101302@edah.org.tw (Y.-C.W.); gn711114@gmail.com (P.-W.T.)

**Keywords:** ^99m^Tc-TRODAT-1, Parkinson’s disease, support vector machine, logistic regression

## Abstract

The neuroimaging techniques such as dopaminergic imaging using Single Photon Emission Computed Tomography (SPECT) with ^99m^Tc-TRODAT-1 have been employed to detect the stages of Parkinson’s disease (PD). In this retrospective study, a total of 202 ^99m^Tc-TRODAT-1 SPECT imaging were collected. All of the PD patient cases were separated into mild (HYS Stage 1 to Stage 3) and severe (HYS Stage 4 and Stage 5) PD, according to the Hoehn and Yahr Scale (HYS) standard. A three-dimensional method was used to estimate six features of activity distribution and striatal activity volume in the images. These features were skewness, kurtosis, Cyhelsky’s skewness coefficient, Pearson’s median skewness, dopamine transporter activity volume, and dopamine transporter activity maximum. Finally, the data were modeled using logistic regression (LR) and support vector machine (SVM) for PD classification. The results showed that SVM classifier method produced a higher accuracy than LR. The sensitivity, specificity, PPV, NPV, accuracy, and AUC with SVM method were 0.82, 1.00, 0.84, 0.67, 0.83, and 0.85, respectively. Additionally, the Kappa value was shown to reach 0.68. This claimed that the SVM-based model could provide further reference for PD stage classification in medical diagnosis. In the future, more healthy cases will be expected to clarify the false positive rate in this classification model.

## 1. Introduction

All of our everyday human behaviors and actions are associated with the transmission of neural signals. Neurotransmission can be broadly classified into electrical transmission and chemical transmission. Electrical transmission is achieved by establishing a link between two neighboring cells. This form of transmission is rapid and bidirectional. It is often observed in human smooth muscle and myocardial cells. Chemical transmission is the transmission of specific chemicals, known as neurotransmitters, secreted by presynaptic neurons to neurotransmitter receptor proteins on the postsynaptic neuron membrane. When the neurotransmitters bind with the receptors, the postsynaptic neurons are either activated or inhibited [1]. Although chemical transmission is unidirectional and slower than electrical transmission, more effector cells can be influenced at the same time. Chemical neurotransmission is extremely active in the central nervous system (CNS) and peripheral nervous system (PNS). It is the most common form of neurotransmission in the human body [2].

Dopamine and acetylcholine are two of the neurotransmitter chemicals that help cells transmit impulses. They are responsible for the transmission of emotional and sensory information across neural pathways. A balance between dopamine and acetylcholine is strictly maintained in the neurotransmission system [3], and their imbalance can lead to neurotransmitter-related diseases. The motor symptoms of Parkinson’s Disease (PD) are due to the fact that the death of cells in the substantia nigra results in insufficient secretion of dopamine [4]. Insufficient dopamine secretion leads to an overabundance of acetylcholine, which triggers resting limb tremors, slowness of voluntary movement, and rigidity. Physical symptoms of PD can be classified into different stages using the Hoehn and Yahr Scale (HYS). The HYS was initially proposed in the Journal of Neurology in 1967 [5]. The Scale relies on physical examination to assess the stage of the disease. Currently, the HYS is still considered the diagnostic reference standard in clinical practice [6].

Although PD is a common degenerative disease, there are still diagnostic difficulties in patients with very early stages of the disease or in elderly patients. The drug challenge test is applied in clinical practice to determine pathological properties. The test involves taking synthetic drugs, such as levodopa, dopamine agonists, or anticholinergics, to increase dopamine levels. This allows physicians to observe and determine whether patients are responsive to the drugs and to distinguish between PD and PD-like syndromes [7]. Besides the physical examination and drug challenge test, non-invasive imaging also provides a valuable diagnostic reference to diagnose PD [8].

Nuclear medicine entails administering drugs containing radioactive tracers and then using nuclear imaging instruments to detect the distribution of the tracers in the body and obtain a visual representation of the internal physiological functions [9]. This functional imaging approach can be used to quantitatively calculate the physiological parameters of different organs, such as the glomerular filtration rate (GFR), effective renal plasma flow (ERPF), and organ uptake ratio. These parameters cannot be determined with the naked eye. However, they can be cross-referenced to determine abnormal organ functioning. Currently, there are many tracers for the dopamine system, including 18F-DOPA–a labeled precursor of dopamine, ^99m^Tc-TRODAT-1 and 123I-β-CIT–imaging agents of the transporter responsible for recycling released dopamine, and 11C-PHNO and 123I-IBZM–radioligands for the postganglionic receptors of dopamine. At present, ^99m^Tc-TRODAT-1 single photon emission computed tomography (SPECT) is the most widely applied approach in Taiwan.

^99m^Tc-TRODAT-1 tracers mark the dopamine transporter (DAT) in the neuronal presynaptic membrane by binding to the relevant receptors after intravenous injection. Then, SPECT is used to evaluate the degree of radioactivity, indirectly indicating the function and the number of dopamine neurons in the striatal pathways [10]. Several earlier studies used the region of interest (ROI) ratio to quantitatively calculate the specific uptake ratio in the striatum and assess brain neurotransmitter-associated diseases, thereby achieving diagnostic objectives [11,12,13,14,15,16]. However, observing the striatal uptake ratio by the selection of the ROI region may produce subjective errors. In addition, these studies did not discuss the validity of PD tests.

The purpose of this study was to calculate whole-brain activity distribution and determine the imaging features of striatal activity volume, followed by analyzing and extracting distinctive features to serve as explanatory variables for modeling, and then using different classification methods to establish a SPECT PD classification model. This model developed from whole-brain activity distribution and striatal activity volume will benefit physicians in the diagnosis of PD and help with long-term follow-up treatment outcomes.

## 2. Materials and Methodology

### 2.1. Subjects

We collected the ^99m^Tc-TRODAT-1 imaging and diagnostic reports archived in the Picture Archiving and Communication System (PACS) between March 2006 and August 2013. According to the diagnostic reports from nuclear medicine specialists, six of the patients had healthy brain function (three men and three women). These patients were between the ages of 31 and 70, with a median age of 47.5 years. There were 196 patients with PD (80 men and 116 women), whose ages were between 25 and 91, with a median age of 69.13 years (Figure 1). Cases in which patients received an overall ^99m^Tc-TRODAT-1 dose of 25 to 30 millicuries (mCi) and then underwent imaging within 2.5 to 4 h after administration were included in the study. However, patients who experienced head tremors were excluded, along with patients who received drug treatment during imaging since interference of the drugs with the efficacy of ^99m^Tc-TRODAT-1 would make outcomes unreliable. Therefore, a total of 202 ^99m^Tc-TRODAT-1 imaging and diagnostic reports were examined in this study. This clinical study was approved by the Medical Ethics Committee of E-DA Hospital. All patients signed an informed consent form prior to participation.

### 2.2. Imaging Instrument and Criteria

A dual-head SPECT instrument (E.cam^TM^ Signature Series Fixed 180; Siemens Medical Solutions Inc.) equipped with the Siemens E.soft Workstation and a fan beam collimator was employed in this study. The field of view (FOV) of a single detection head was 53.3 × 38.7 cm^2^, and the diagonal field of view (Diagonal FOV) was 63.5 cm. A single detection head is equipped with 59 photomultiplier tubes in a hexagonal arrangement and uses 5/8 inch sodium iodide (NaI(Tl)) crystals with a crystal size of 59 × 44.5 cm^2^. A step-by-step scan method was adopted for performing DAT scan using ^99m^Tc-TRODAT-1. The image matrix was 128 × 128. A total of 64 images were captured at a collection rate of 25 s per image. Filtered back projection (FBP) was adopted for image reconstruction. The filter was a low-pass Butterworth filter with a cutoff frequency of 0.4 and an order of 8.

### 2.3. Experimental Design

This experimental is a retrospective study. A total of 202 ^99^mTc-TRODAT-1 SPECT imaging and diagnostic reports were collected. Among these were six healthy patients, 102 patients with mild PD (HYS Stage 1 to Stage 3), and 94 patients with severe PD (HYS Stage 4 and Stage 5) [5,17,18]. A three-dimensional method was used to estimate the features, including skewness, kurtosis, Cyhelsky’s skewness coefficient, Pearson’s median skewness, DAT activity volume, and DAT activity maximum, in the images of activity distribution and striatal activity volume [14,19,20]. Subsequently, the reports were randomly allocated into two groups, each containing 101 reports. One group served as the training group, and the other served as the validation group. Then, significant statistical measures of the features was identified using descriptive statistics, the nonparametric Mann–Whitney U test, the Kruskal–Wallis test, and the Dunn–Bonferroni multiple comparison test. The outcome data were modeled using logistic regression (LR) and support vector machine (SVM) classification (n = 101) [18,21,22,23]. Finally, we developed a PD classification model for SPECT (n = 101) (Figure 2).

We adopted two approaches for selecting the ROI during three-dimensional evaluation, namely, the single threshold method and the seed region growing method [24]. The single threshold method used the whole-brain as the ROI in the reconstructed ^99m^Tc-TRODAT-1 SPECT image to determine the intensity count value within the region. In nuclear imaging, the intensity contrast derived from the count value of the image. The single threshold method effectively eliminated the background count value from the ROI, reducing the calculation error. We set the single threshold value at 15 (because 15 was the maximum value of the axial section located on the periphery of the brain). Any count value higher than 15 denoted its inclusion in the ROI covering the entire brain. Image histograms were plotted to highlight the differences in the morphological distributions between the healthy patients and patients with various stages of PD (Figure 3). Finally, we quantified the patterns of the histograms. The resulted quantitative statistical measures described the brain activity distributions of the patients.

The seed region growing method categorized the image data by sampling the greyscale values, color values, and intensities of neighboring pixels of initial seed points. We used the seed region growing method to calculate the volume and region of striatal activity (Figure 4) [25]. The initial region-growing value was the maximum count value within the axial section of the striatum, therefore, the maximum striatal activity volume was able to be obtained by this method. The resulted size of the growth region could be used to explain the features of striatal activity volume.

There were six types of statistical measures adopted in this study. They were the outcomes calculated and extracted from the two methods mentioned above, namely single threshold method which gave skewness (SK), kurtosis (KUR), Cyhelsky’s skewness coefficient (CSK), and Pearson’s median skewness (MES) in the ROI covering the whole-brain, and seed region growing method which gave Dopamine Transporter Activity Volume (DTAV) and Dopamine Transporter Activity Maximum (DTAM). The aforementioned statistical measures included the measurement of whole-brain activity distribution (SK, KUR, CSK, and MES) and striatal activity volume (DTAV and DTAM). The calculation methods for the various statistical measures are shown in Table 1. Skewness was used to measure the skewness of the activity distribution by comparing the whole-brain count value to the whole-brain mean count value. Kurtosis measured the activity distribution curve by comparing the whole-brain count value and the whole-brain mean count value, where xi represented the count value of the *ith* pixel in the image, $\bar{x}$ represented the mean count value of the image, and n represented the number of pixels selected in the image. The CSK described the skewness of the activity distribution using the number of pixels with count value greater than or less than the mean count value, where n_below_ represented the number of pixels with count value less than the mean count value, n_above_ represented the number of pixels with count value greater than the mean count value, and n_total_, the total number of pixels in the selected images. MES measured the skewness of the activity distribution by comparing the whole-brain median count value and the whole-brain mean count value $\bar{x}$ was the mean of the count values of all selected images. *Md* was the median of the count values of all selected images, and *SD* was the standard deviation of the count values of all selected images.

Previous studies largely focused on analyzing the striatal activity region. However, the histograms of these studies showed different activity distributions between the normal parts of the brain and the parts affected by PD. Therefore, we incorporated the quantified statistical measures of the histograms to describe the features of whole-brain activity distribution. Then, descriptive statistics, nonparametric Mann–Whitney U test, Kruskal–Wallis test, and Dunn–Bonferroni multiple comparison test were performed to determine the features of the statistical measures associated with whole-brain activity distribution and striatal activity volume, select the statistically significant features, and thereby create a classification model for PD patients. 

### 2.4. Classification Implementation

The logistic regression (LR) model is one of the discrete selection method models. It belongs to the category of multivariate analysis and is widely used in biostatistics, medicine, econometrics, marketing, and so on. The logistic regression model is used to analyze and interpret the relationship between a nominal scale dependent variable and independent variable. The binary logistic regression model can be derived from the linear probability model (LPM) equation. In a logistic regression model, such as Equations (1) and (2), where *P(X)* is the probability of an event affected by factor X, the *x*_1_, *x*_2_, …, *x*_k_ is an independent variable, such as body weight, age, or image feature; the *β*_0_, *β*_1_, …, *β*_K_ are the estimated parameters. The general regression analysis is to explore the relationship between two or more variables and to establish the mathematical function model to observe the correlation between the independent variables and the reaction variables, such as success or failure, normal, or abnormal, etc.
(1)$$P(X)=\frac{e^{f(x)}}{1+e^{f(x)}},\;0\leq P(X)\leq 1$$
(2)$$f(x)=\beta_{0}+\beta_{1}x_{1}+\beta_{2}x_{2}+\ldots +\beta_{k}x_{k}$$

The SVM is widely used in statistical classification. It is a supervised learning method. In training data, the SVM is to find the hyper-plane in each category. The data on the hyper-plane, which constitutes a support vector with sufficient information, can perform effective image classification with less training samples. There are various kernels that can be used during SVM training, including linear, quadratic, polynomial, and radial basis function (RBF) kernels. SVM was performed using the WEKA (https://www.cs.waikato.ac.nz/ml/weka/index.html) software. It can be used to handle data unbalanced situation. In this study, SVM with RBF kernel was used. Finally, the classification is performed using the SVM or logistic regression and evaluated with the percentage split (50% of the data formed the training set for building the model, while the other 50% formed the test set for testing the model) validation strategy.

## 3. Results

The descriptive statistics of the six statistical measures based on three disease stage categories—healthy, mild (HYS Stage 1 to Stage 3), and severe (HYS Stage 4 and Stage 5)—were tabulated in Table 2. Descriptive statistics included mean, 95% confidence interval (upper and lower bounds), standard deviation value, minimum value, and maximum value. Statistical results indicated that the mean and standard deviation of skewness in the healthy, mild, and severe groups were 0.52 ± 0.49, 0.05 ± 0.29, and −0.21 ± 0.26, respectively. These findings indicated that the activity distribution of the healthy subjects was right-skewed, that of the patients with mild PD approximated normal distribution, and that of the patients with severe PD was left-skewed. The kurtosis values for the healthy, mild, and severe groups were 4.98 ± 1.52, 3.27 ± 0.56, and 2.67 ± 0.33, respectively. The findings indicated that the activity distribution curves of the healthy subjects and patients with mild or severe PD all formed leptokurtic curves. A comparison of the three groups revealed that the leptokurtic curves of the healthy patients were the most obvious. The Cyhelsky’s skewness values for the healthy, mild, and severe groups were −0.01 ± 0.09, −0.05 ± 0.09, and −0.07 ± 0.08, respectively. The findings revealed that the activity distributions in all three groups were left-skewed. However, skewness was most obvious in the severe PD group. The Pearson’s median skewness values for the healthy, mild, and severe groups were −0.04 ± 0.28, −0.17 ± 0.31, and −0.26 ± 0.27, respectively. Similarly, the activity distributions in all three groups were left-skewed, and skewness was most obvious in the severe PD group. The DTAV of the healthy, mild, and severe groups was shown to be 33.68 ± 6.58, 15.85 ± 4.63, and 10.09 ± 4.43, respectively. The findings revealed that the DTAV of the healthy patients was higher than that of the PD patients, and the discrepancy became greater as PD progressed. The DTAM of the healthy, mild, and severe groups were 291.33 ± 137.53, 364.9 ± 89.83, and 298.24 ± 85.84, respectively. The findings revealed that the DTAM of the mild PD patients were higher than that of the severe PD patients. Due to the slight differences among the three groups in several statistical measures, we subsequently performed Kruskal–Wallis test and Dunn–-Bonferroni multiple comparison test to evaluate the differences between the groups. Descriptive statistics explained the concentration and dispersion of the six statistical measures among the various classifications. However, they could not explain whether the statistical measures were significantly different among the groups. Therefore, the Kruskal–Wallis test and multiple comparison analysis tests were performed to determine the differences between the groups. A *p-value* of 0.05 was chosen as its statistical significance cut-off.

In this study, we used the Kruskal–Wallis test to examine whether there were significant differences in the data of the three groups and compare such differences. Table 2 showed the variance of the six statistical measures extracted from the images of the healthy, mild PD, and severe PD patients. Results indicated that the differences in the three groups of patients (healthy, mild, and severe) regarding the six statistical measures achieved statistical significance. The p-values for SK, KUR, CSK, MES, DTAV, and DTAM were *p* < 0.001, *p* < 0.001, *p* = 0.046, *p* = 0.033, *p* < 0.001, and *p* < 0.001, respectively. However, the Kruskal–Wallis test results only highlighted significant differences between the three patient groups. They did not show detailed differences between individual patients. Therefore, multiple comparison test was performed to determine differences between individual patients.

In this study, the Dunn–Bonferroni multiple comparison test was adopted for multiple comparison testing. The results of the Dunn–Bonferroni multiple comparison test were shown in Table 3. The table showed that KUR and DTAV achieved significant statistical differences (*p* < 0.05). SK was similar in the healthy patients and patients with mild PD. However, the difference in SK in the healthy patients and those with severe PD achieved statistical significance (*p* < 0.05), suggesting that SK was a key statistical measure in this study. The difference in DTAM between the patients with mild and severe PD achieved statistical significance (*p* < 0.05). However, there was no statistically significant difference in the DTAM in the healthy patients and those with mild or severe PD. Finally, CSK and MES were similar among the three groups. These results showed that SK, KUR, and DTAV were key statistical measures for distinguishing between healthy patients, patients with mild PD, and patients with severe PD. Therefore, these three statistical measures were adopted as explanatory variables for PD classification. Subsequently, we developed PD classification models using logistic regression and SVM (n = 101).

Regarding the classification, we first developed independent models based on SK, KUR, and DTAV, respectively. Then, we also created a model using both SK and KUR and another using SK, KUR, and DTAV all together, thereby a total of five classification variable groups. To clearly explain the classification results for each of the variable groups, we redefined DTAV as the feature of activity volume (FAV), SK and KUR as the features of activity distribution (FAD), and SK, KUR, and DTAV as the features of activity distribution and volume (FADV; Table 4). Finally, the LR and SVM classification methods were used to convert the five groups of explanatory variables into models. The validity of the models was analyzed to identify the optimal classification model.

The validity of the disease classification models created with the LR and SVM methods was tested by calculating sensitivity, specificity, positive predictive value (PPV), negative predictive value (NPV), and area under curve (AUC). Among the models, the FADV model created with the SVM classification method achieved the highest validity. Its sensitivity, specificity, PPV, NPV, accuracy, and AUC were 82.8%, 100%, 83.7%, 66.7%, 83.2%, and 84.5%, respectively, suggesting that the model achieved excellent discrimination. The classification consistency of the model (Kappa value) was 0.68, suggesting that the report results are highly substantial. The validity with respect to the various classification variables modeled with the LR and SVM method was shown in Table 5. 

## 4. Conclusions

In this study, the sample size of healthy subjects was relatively small because of the retrospective experimental design. We used a graphical approach to analyze ^99m^Tc-TRODAT-1 SPECT images and determine whole-brain activity distribution and striatal activity volume. A three-dimensional evaluation method was used to estimate relevant statistical measures. Skewness and kurtosis were used to quantify whole-brain activity distribution curves. DTAV was used to quantify striatum activity volume. A PD (healthy, mild, and severe) classification model was created by modeling FADV (including SK, KUR, and DTAV) with the SVM classification method, and an accuracy of 83.2% was achieved. Of the FADV classification variable group, the accuracy of the SVM model in distinguishing between healthy patients, patients with mild PD, and patients with severe PD was 1% higher than that of the LR model, and the Kappa value of the SVM model was 2% higher than that of the LR model. Therefore, the SVM classification model achieved high validity. Furthermore, taking account of whole-brain activity distribution and striatal activity volume facilitates long-term follow-up examination and treatment efficacy. With this model, clinical physicians can have another reference for the classification of PD stages, rather than making subjective judgments only.

^99m^Tc-TRODAT-1 SPECT has been proven to be an effective approach for assessing PD. Previous studies on dopaminergic imaging often applied computed tomography (CT) or magnetic resonance imaging (MRI) for the fusion and positioning of anatomical images. As a result, they focused on accurately selecting the ROI. The analysis approaches conducted in these studies were largely based on quantifying striatal uptake ratio by examining the basal nucleus and background regions (cerebellum or frontal lobe). The results of most of these studies showed that the striatal uptake ratio in normal regions of the brain was higher than that in the regions affected by PD and decreased concurrently with the increase in PD severity and stages. The ROI ratio analysis method could be used to observe striatal uptake ratio and further test and classify PD.

In this study, we adopted a graphical approach to analyze ^99m^Tc-TRODAT-1 SPECT images, taking into account the distribution of grayscale values and the range of activity within the striatum. A three-dimensional evaluation method was used to calculate skewness, kurtosis, and DTAV. The kurtosis and DTAV values of the various groups achieved significant statistical differences (p < 0.05). Multiple comparison results indicated that skewness was similar in healthy patients and those with mild PD (p > 0.05). However, skewness in healthy subjects was significantly different from that in severe PD patients (p < 0.05). Therefore, SK was a key statistical measure in this study. We also found that the activity distribution of healthy patients was right-skewed, that of mild PD patients approximated normal distribution, and that of severe PD patients was left-skewed.

## 5. Research Limitations and Future Work

A limitation of this study was the use of a retrospective study design. This limited the inclusion criteria of healthy subjects. Therefore, the sample size was relatively small. In the future, the number of TRODAT-1 SPECT cases are expected to be increased. In addition, MRI and CT anatomical images can be included in ROI designs as a reference. The statistical features can consider using probability distribution function (PDF) method. That is more general and might be more suitable for modeling real-world problems. Researchers can also attempt to use other classification methods to create models for the indication of PD stages and establish AI or deep-learning classification models for other neurotransmitter-related diseases to identify the optimal classification models for such diseases.

**Ethical Approval:** The Institutional Internal Review Board of E-DA Hospital approved this retrospective experimental study. The approval number is EMRP-100-054.

## Figures and Tables

**Figure 1 sensors-19-01740-f001:**
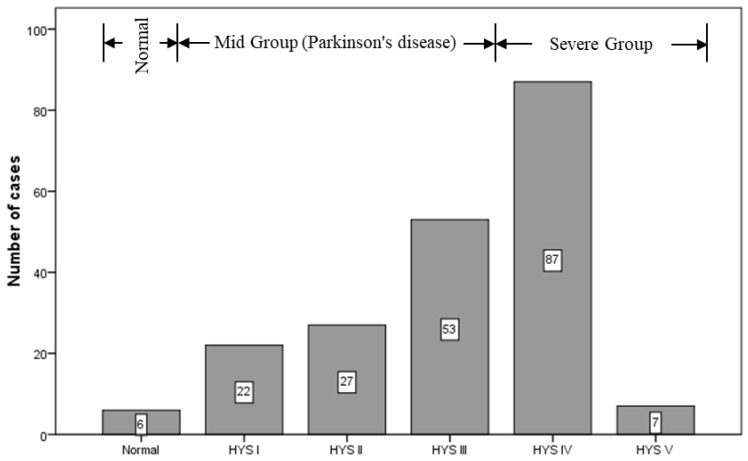
Number of cases according to Hoehn and Yahr Scale (HYS) standard.

**Figure 2 sensors-19-01740-f002:**
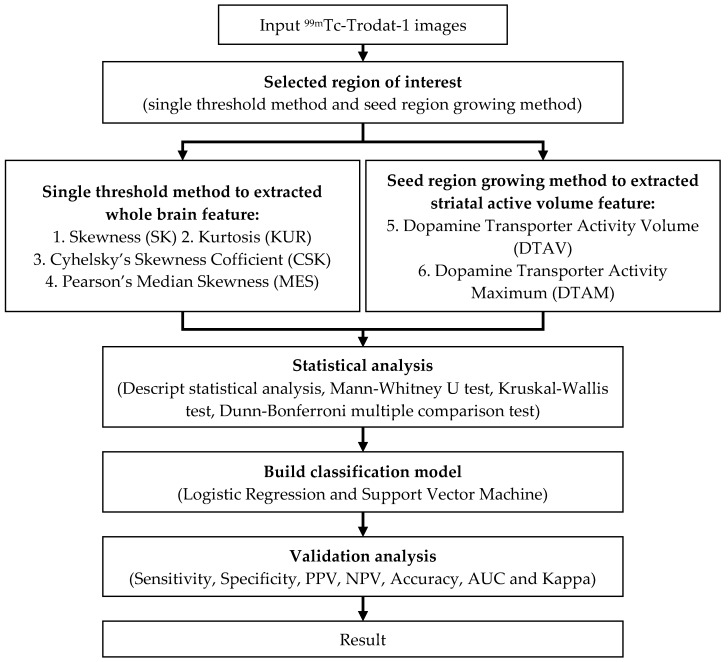
A flow chart of experimental design.

**Figure 3 sensors-19-01740-f003:**
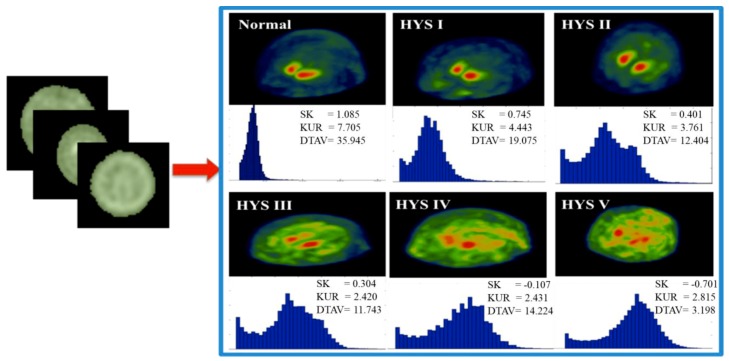
Histogram plots between normal and Parkinson’s Disease (PD) stage. Maximum intensity projection (MIP) shown the calculation of whole brain and correspond to histogram. The histogram can describe active uptake in whole brain via values of skewness (SK).

**Figure 4 sensors-19-01740-f004:**
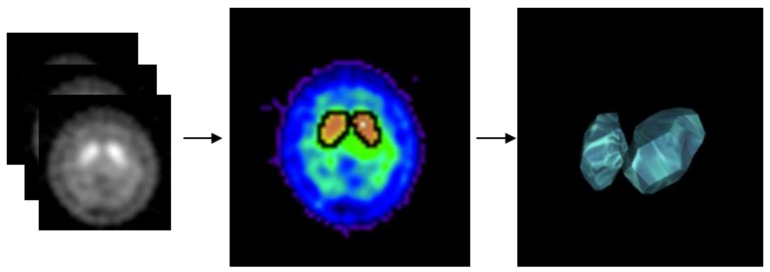
Using Seed region growing method calculate the volume of striatal activity. (**left**) ^99m^Tc-TRODAT-1 SPECT image, (**middle**) striatal activity in single slide, (**right**) whole brain (3D) striatal activity.

**Table 1 sensors-19-01740-t001:** Six features were calculated from whole brain and striatal.

Features	Formula	Location
SK	$Skewness=\frac{\sum {\left(x_{i}-\bar{x}\right)}^{3}}{n-1}{\left(\frac{n-1}{\sum {\left(x_{i}-\bar{x}\right)}^{2}}\right)}^{32}$	Whole-brain
KUR	$Kurtosis=\frac{\sum {\left(x_{i}-\bar{x}\right)}^{4}}{n-1}{\left(\frac{n-1}{\sum {\left(x_{i}-\bar{x}\right)}^{2}}\right)}^{2}-3$	Whole-brain
CSK	$CSK=\frac{n_{below}-n_{above}}{n_{total}}$	Whole-brain
MES	$MES=\frac{3(\bar{x}-Md)}{SD}$	Whole-brain
DTAV	DTAV=∑(Activity area × Single slice thickness)	striatal
DTAM	DTAM=max(Activity area × Single slice thickness)	striatal

**Table 2 sensors-19-01740-t002:** The descriptive statistical and Kruskal–Wallis test results between healthy, mild (HYS Stage 1 to Stage 3), and severe (HYS Stage 4 and Stage 5) (n = 202).

Features	Group	Cases	Mean	95% Confidence Index		Standard Deviation	Minimum	Maximum	Kruskal–Wallis Test (P-value)
				Lower	Upper				
SK	Healthy	6	0.520	0.003	1.041	0.490	−0.330	1.090	<0.001
	Mild	102	0.050	−0.009	0.108	0.290	−0.550	0.750	
	Severe	94	−0.210	−0.263	−0.157	0.260	−0.870	0.380	
KUR	Healthy	6	4.980	3.383	6.574	1.520	3.530	7.710	<0.001
	Mild	102	3.270	3.155	3.375	0.560	2.390	5.040	
	Severe	94	2.670	2.602	2.736	0.330	2.100	4.050	
CSK	Healthy	6	−0.010	−0.111	0.087	0.090	−0.200	0.070	0.046
	Mild	102	−0.050	−0.068	−0.034	0.090	−0.260	0.130	
	Severe	94	−0.070	−0.091	−0.058	0.080	−0.260	0.110	
MES	Healthy	6	−0.040	−0.334	0.256	0.280	−0.550	0.220	0.033
	Mild	102	−0.170	−0.233	−0.112	0.310	−0.890	0.570	
	Severe	94	−0.260	−0.316	−0.205	0.270	−0.810	0.440	
DTAV	Healthy	6	33.680	26.780	40.582	6.580	24.290	40.470	<0.001
	Mild	102	15.850	14.940	16.753	4.630	7.390	28.940	
	Severe	94	10.090	9.178	10.993	4.430	0.720	17.640	
DTAM	Healthy	6	291.330	147.010	435.660	137.530	95.000	459.000	<0.001
	Mild	102	364.900	347.280	382.570	89.830	230.000	739.000	
	Severe	94	298.240	280.660	315.830	85.840	141.000	509.000	

**Table 3 sensors-19-01740-t003:** The Dunn–Bonferroni test results between healthy, mild (HYS Stage 1 to Stage 3), and severe (HYS Stage 4 and Stage 5) (n = 202).

Feature	Control Group	Compare Group	Dunn–Bonferroni Test (P-value)	Feature	Control Group	Compare Group	Dunn–Bonferroni Test(P-value)
SK	Healthy	Mild	0.170	MES	Healthy	Mild	0.704
		Severe	0.001			Severe	0.172
	Mild	Healthy	0.170		Mild	Healthy	0.704
		Severe	0.001			Severe	0.106
	Severe	Healthy	0.001		Severe	Healthy	0.172
		Mild	0.001			Mild	0.106
KUR	Healthy	Mild	0.038	DTAV	Healthy	Mild	0.012
		Severe	0.001			Severe	0.001
	Mild	Healthy	0.038		Mild	Healthy	0.012
		Severe	0.001			Severe	0.001
	Severe	Healthy	0.001		Severe	Healthy	0.001
		Mild	0.001			Mild	0.001
CSK	Healthy	Mild	0.553	DTAM	Healthy	Mild	0.596
		Severe	0.153			Severe	0.999
	Mild	Healthy	0.553		Mild	Healthy	0.596
		Severe	0.193			Severe	0.001
	Severe	Healthy	0.153		Severe	Healthy	0.999
		Mild	0.193			Mild	0.001

**Table 4 sensors-19-01740-t004:** Names of each classified variable.

Item	Group 1	Group 2	Group 3	Group 4	Group 5
Variable	SK	KUR	DTAV	SK, KUR	SK, KUR, DTAV
Name			FAV	FAD	FADV

Annotation: FADV = Features of Activity Volume, FAD = Features of Activity Distribution, FADV = Features of Activity Distribution and Volume

**Table 5 sensors-19-01740-t005:** The validation of classify model between LR and SVM in healthy, mild, and severe patient (n = 101).

Validation	LR					SVM				
	SK	KUR	FAV	FAD	FADV	SK	KUR	FAV	FAD	FADV
Sensitivity	0.667	0.737	0.747	0.768	0.818	0.657	0.747	0.747	0.788	0.828
Specificity	0.500	0.500	1.000	0.500	1.000	0.500	0.500	1.000	0.500	1.000
PPV	0.660	0.730	0.747	0.760	0.827	0.650	0.755	0.747	0.780	0.837
NPV	1.000	1.000	1.000	1.000	0.667	1.000	0.333	1.000	1.000	0.667
Accuracy	0.663	0.733	0.752	0.762	0.822	0.653	0.753	0.752	0.782	0.832
AUC	0.738	0.840	0.845	0.866	0.905	0.739	0.790	0.768	0.865	0.845
Kappa	0.344	0.482	0.527	0.539	0.661	0.326	0.509	0.523	0.578	0.680

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
