# Peer review of "Feasible Classified Models for Parkinson Disease from 99mTc-TRODAT-1 SPECT Imaging"

_sensors, 2019, doi:10.3390/s19071740_

Round 1

Reviewer 1 Report

The manuscript by Shih-Yen Hsu and colleagues shows a try to develop a new concept to classify subjects with Parkinson’s disease at different stages using 99mTc-TRODAT-1 SPECT Imaging data. A total of 202 99mTc-TRODAT-1 SPECT imaging were used in this study. PD patient were separated into two categorize; mild and advanced PD according to the Hoehn and Yahr Scale standard. Statistical and voxels intensity related features were extracted from data. Two ML models were used i.e. logistic regression and support vector machine were used for the classification.  Errors in statistical formulas raise serious concerns about the data’s validity fed into the ML models. This makes it basically hard to estimate the value of this manuscript. My main concerns revolve around the features’ extraction. The novelty of the work is limited to the choice of features because the technique of classification per se is classic and was applied on SPECT/PET data in several published works. However, there are still questions remain to be addressed as detailed below.

Major comments:

1-     The clinical goal behind the use of 99mTc-TRODAT-1 SPECT imaging is to give a guide to the attenuation of functional dopaminergic neurons in the nigrostriatal circuit. Including the activity of the whole-brain activity is meaningless because the non-specific binding response conceals the specific binding. Therefore, the distribution informs more about non-specific binding rather than specific-binding. Authors might use only the striatum ROI only to derive accurate features.

2-     The kurtosis formula in Table1 is wrong: kurtosis is the fourth moment divided by squared second moment. 

3-     Pearson’s Median Skewness defined in Table1 is wrong. There is no power in the PMS, it is 3 times the distance between the entry and its median normalized by the sample size. I am wondering if it is just a typo mistake or authors used these formulas to calculate Kurtosis and PMS!

Author Response

Response to Reviewer 1 Comments 

Point 1: The clinical goal behind the use of 99mTc-TRODAT-1 SPECT imaging is to give a guide to the attenuation of functional dopaminergic neurons in the nigrostriatal circuit. Including the activity of the whole-brain activity is meaningless because the non-specific binding response conceals the specific binding. Therefore, the distribution informs more about non-specific binding rather than specific-binding. Authors might use only the striatum ROI only to derive accurate features.

Response 1:  Thanks for reviewer suggestion. In the literature, most of the features identify of dopaminergic focused on analyzing the striatal activity region. In this study, we not only determine the imaging features of striatal activity volume and also calculate whole-brain activity distribution. In the results, a PD (healthy, mild, and severe) classification model from features SK, KUR, and DTAV with the SVM classification method, the accuracy was 83.2%. If only used striatal activity volume to classify the accuracy was 75.2%.

Point 2:    The kurtosis formula in Table1 is wrong: kurtosis is the fourth moment divided by squared second moment.

Response 2:  Thanks for reviewer remind. The kurtosis formula was corrected.

Point 3:     Pearson’s Median Skewness defined in Table1 is wrong. There is no power in the PMS, it is 3 times the distance between the entry and its median normalized by the sample size. I am wondering if it is just a typo mistake or authors used these formulas to calculate Kurtosis and PMS!

Response 3: Thanks for reviewer remind. The Pearson’s Median Skewness formula was corrected. And we sure this is a typo mistake. In this work, all feature values were calculated by the mathematical toolbox from the MATLAB program.

Reviewer 2 Report

Feasible Classified Models for Parkinson Disease from 99mTc-TRODAT-1 SPECT Imaging

The study is interesting and well stated. However, I have some comments to this study as follow: 

1- The authors should provide more details related to the pre-processing section.

2- More information about the machine learning part is needed. Especially, the validation procedure is not clear in this study. Which kernel is used for SVM and how to tune the parameters. 

3- The groups are unbalanced. SVM has no good performance with unbalanced data. the authors should clarify how to deal with this point. 

4- In the Fig.2, the author provided the histogram from each group. The histogram procedure has been used successfully in some AD detection studies. see the following studies as an example.

I suggest the authors to apply that procedure (histogram-based feature generation through brain imaging data) their data and provide a comparison to with this statistical features. 

"Probability distribution function-based classification of structural MRI for the detection of Alzheimer’s disease." Computers in biology and medicine 64 (2015): 208-216.

"Histogram-based feature extraction from individual gray matter similarity-matrix for Alzheimer’s disease classification." Journal of Alzheimer's Disease 55.4 (2017): 1571-1582.

Author Response

Response to Reviewer 2    Comments

Point 1: The authors should provide more details related to the pre-processing section.

Response 1: Thanks for reviewer comment. We added a flow chart of experimental design and provide more details about imaging instrument in the section 2.2.

Point 2: More information about the machine learning part is needed. Especially, the validation procedure is not clear in this study. Which kernel is used for SVM and how to tune the parameters.

Response 2: Thanks for reviewer comment. In additional section 2.4 wrote more information about the machine learning part. The validation of SVM and LR were utilize percentage split (50% of the data formed the training set for building the model, 50% of the data formed the test set for testing the model) strategy. In this study, SVM with RBF kernel was used.

Point 3: The groups are unbalanced. SVM has no good performance with unbalanced data. the authors should clarify how to deal with this point.

Response 3: In literature, there have been developed at least two kind of method (pre-processing and algorithmic modification) to solve the unbalanced group situation for SVM. There have been several techniques proposed in the literature to make the SVM algorithm less sensitive to the class imbalance by modifying the associated kernel function. In this study, the normalized polynomial kernel, polynomial kernel, function-based universal kernel, radial basis function kernel and subsequence kernel were used in classify via Weka software. Final, SVM with RBF kernel was used in this study.

Point 4: In the Fig.2, the author provided the histogram from each group. The histogram procedure has been used successfully in some AD detection studies. see the following studies as an example.

I suggest the authors to apply that procedure (histogram-based feature generation through brain imaging data) their data and provide a comparison to with this statistical features. 

"Probability distribution function-based classification of structural MRI for the detection of Alzheimer’s disease." Computers in biology and medicine 64 (2015): 208-216.

"Histogram-based feature extraction from individual gray matter similarity-matrix for Alzheimer’s disease classification." Journal of Alzheimer's Disease 55.4 (2017): 1571-1582.

Response 4: Thanks for reviewer suggestion. In the Fig.2, we have shown the statistical features (SK, KUR, DTAV) to provide a comparison with different stage of PD.

Round 2

Reviewer 1 Report

Thank the authors for their careful reply and explanations. The revised manuscript is much clear with adding the section 2.4.  I assume that the errors in the statistical formulas were just a typo error since the authors kept the same results.  No other comments.

Author Response

Response to Reviewer 1 Comments

Point 1: Thank the authors for their careful reply and explanations. The revised manuscript is much clear with adding the section 2.4.  I assume that the errors in the statistical formulas were just a typo error since the authors kept the same results.  No other comments.

Response 1: Thanks for reviewer kindly comment.

Reviewer 2 Report

        With this revision, the authors have addressed some of my previous concerns. However, the utility of the proposed method would be better supported by addressing the following concerns.  

1- To validate the proposed method, the author used 50% data as training set to build a model and 50% as test.  How do the authors separate the data? Statistically, reporting the results based on only one model is not robust.  For more valid evaluation, the authors should randomly select the training and test sets (50% data as training, 50% as test) and repeat this procedure at least 100 times. Indeed, a more robust evaluation result is needed. 

2- In the response to reviewer, the authors stated that " (preprocessing and algorithmic modification) to solve the unbalanced group situation for SVM." It is unclear for me how the preprocessing step can be useful to handel data imbalanced in SVM!! please clarity this point. 

3-In the response to reviewer, the author stated that “In this study, the normalized polynomial kernel, polynomial kernel, function-based universal kernel, radial basis function kernel and subsequence kernel were used in classify via Weka software. Final, SVM with RBF kernel was used in this study.”. I am not familiar with Weka software. Does this software provide the techniques to handel data imbalanced in SVM!! The authors need to address this point in the manuscript.  

4- Again, I have a concern about data imbalanced in SVM. However, this point is addressed in the limitation.  BTW, it is suggested to put the research limitation before conclusion.  

5- it was suggested to use “"Probability distribution function” of ROIs as statistical features for your proposed. Although, the authors extracted the statistical features (SK, KUR, DTAV), but this method (i.e., PDF of selected regions) could be considered as future work in the paper.

Author Response

Response to Reviewer 2 Comments

With this revision, the authors have addressed some of my previous concerns. However, the utility of the proposed method would be better supported by addressing the following concerns. 

Point 1: To validate the proposed method, the author used 50% data as training set to build a model and 50% as test.  How do the authors separate the data? Statistically, reporting the results based on only one model is not robust.  For more valid evaluation, the authors should randomly select the training and test sets (50% data as training, 50% as test) and repeat this procedure at least 100 times. Indeed, a more robust evaluation result is needed.

Response 1: Thanks for reviewer comment. The data was separated by random number using default Fisher-Yates shuffle method. Then the first 50% of the instances in the shuffled data will be used for training and the rest for testing.

Point 2: In the response to reviewer, the authors stated that " (preprocessing and algorithmic modification) to solve the unbalanced group situation for SVM." It is unclear for me how the preprocessing step can be useful to handel data imbalanced in SVM!! please clarity this point.

Response 2: About the preproscessing of the imbalanced group situation for SVM. There were two ways mention in the literature that is resampling methods and ensemble learning methods. In resampling include random and focused under/oversampling methods and synthetic data generation methods. And ensemble learning has also been applied as a solution for training SVM with imbalanced datasets. The majority class dataset is separated into multiple subdatasets such that each of these sub-datasets has a similar number of examples as the minority class dataset.

Point 3: In the response to reviewer, the author stated that “In this study, the normalized polynomial kernel, polynomial kernel, function-based universal kernel, radial basis function kernel and subsequence kernel were used in classify via Weka software. Final, SVM with RBF kernel was used in this study.”. I am not familiar with Weka software. Does this software provide the techniques to handel data imbalanced in SVM!! The authors need to address this point in the manuscript. 

Response 3: All these kernel functions were provided by the Weka software and it could be used to handle data imbalanced problem in SVM. About this point we were address in the manuscript line 218.

Point 4: Again, I have a concern about data imbalanced in SVM. However, this point is addressed in the limitation.  BTW, it is suggested to put the research limitation before conclusion. 

Response 4: Thanks for reviewer suggestion. We put the research limitation before conclusion in line 300.

Point 5: it was suggested to use “"Probability distribution function” of ROIs as statistical features for your proposed. Although, the authors extracted the statistical features (SK, KUR, DTAV), but this method (i.e., PDF of selected regions) could be considered as future work in the paper.

Response 5: Thanks for reviewer suggestion. We added the suggestion in the section 5 lines 336-338.

Round 3

Reviewer 2 Report

The authors correctly responded to the comments.